# Effect of Agricultural Structure Adjustment on Spatio-Temporal Patterns of Net Anthropogenic Nitrogen Inputs in the Pearl River Basin from 1990 to 2019

Kai Xu [1,2] , Jiaogen Zhou [2,*,†] , Qiuliang Lei [3], Wenbiao Wu [4] and Guangxiong Mao [2]

[1]  School of Chemical Engineering, Northeast Electric Power University, Jilin 132012, China
[2]  Jiangsu Provincial Engineering Research Center for Intelligent Monitoring and Ecological Management of Pond and Reservoir Water Environment, Huaiyin Normal University, Huaian 223300, China
[3]  Key Laboratory of Nonpoint Source Pollution Control, Ministry of Agriculture and Rural Affairs, Institute of Agricultural Resources and Regional Planning, Chinese Academy of Agricultural Sciences, Beijing 100081, China
[4]  Research Center of Information Technology, Beijing Academy of Agriculture and Forestry Sciences, Beijing 100097, China
*   Correspondence: zhoujg@hytc.edu.cn
†   Current address: No.111 West Changjiang Road,  Huai'an 223300, China.

**Abstract:** Worldwide urbanization has brought dramatic changes in agricultural structures, as well as serious agricultural non-point source pollutions of nitrogen and phosphorus. However, understanding the effect of agricultural structure adjustment on net anthropogenic nitrogen inputs (NANI) has been still limited. In this paper, statistical data from the agricultural statistical Yearbook, the National Economic and Social Development Statistical Bulletin were collected from 1990 to 2019 in the Pearl River Basin, China, and used to analyze the spatial and temporal patterns of NANI and its influencing factors. The results indicated that the agricultural structure adjustment has significantly influenced the spatial and temporal patterns of NANI in the last 30 years in the Pearl River Basin. The NANI decreased from 1990 to 2019, and had a spatial pattern of higher values in the upstream areas and lower in the downstream areas. In terms of the nitrogen input sources of NANI, in the economically developed regions downstream, nitrogen inputs are dominated by food/feed nitrogen, which accounted for an average of 49.6% of total nitrogen inputs. In upstream areas with relatively low economic development, fertilizer nitrogen accounted for an average of 54.9% of total nitrogen inputs. A novel nitrogen input source index of NANI, namely the ratio of agricultural nitrogen inputs to non-agricultural nitrogen inputs of NANI(ASNA), was also proposed to characterize the impact of the agricultural industry restructuring on NANI changes over time. Similar to the characteristics of NANI from 1990 to 2019, the ASNA showed a decreasing trend in the study area. Moreover, agricultural variables (agricultural land area, nitrogen fertilizer consumption and livestock farming density) tended to contribute less to the explained ASNA variances, while the contributions of the non-agricultural factors (population density and non-agricultural GDP) increased from 1990 to 2019. This indicated that the contribution of nitrogen inputs from agricultural sources to the NANI decreased while the contribution of nitrogen inputs from non-agricultural sources increased, with the shifts of agricultural sectors to the secondary and tertiary sectors in the Pearl River Basin. Our findings also suggest that differently regional targeting should be considered for the nitrogen pollution management in the Pearl River Basin, which focuses on the nitrogen pollution management of non-agricultural sources in the downstream areas, and but highlights agricultural nitrogen pollution management in the upstream areas.

**Keywords:** net anthropogenic nitrogen inputs (NANI); nitrogen pollution; agricultural non-point source pollution; agricultural structure adjustment; agricultural landuse; nitrogen fertilizer consumption; livestock farming

## 1. Introduction

Nitrogen is an abundant and essential element for crop growth and plays an important role in securing global food production. Global economic development and population growth rely heavily on the current pattern of a large quantities of nitrogen fertilizer inputs to ensure increasing crop yields. Sixteen major crops, including wheat, maize, rice and soybeans, consume 70% of global nitrogen fertilizer inputs [1]. Global crop N use efficiency has decreased by 10% over the last 50 years [2], indicating a significant loss of nitrogen into the ecosystems.This unused nitrogen enters surface water, groundwater and the atmosphere through runoff migration, infiltration and volatilization, and finally triggered environmental problems such as the eutrophication of water bodies, groundwater pollution and air pollution [3–5]. Watersheds are the best geographical unit for agricultural non-point source pollution management [6]. Accounting for watershed-scale nitrogen inputs helps one to develop a precise policy of nitrogen pollution control.

Nitrogen balance models are mainly divided into two types: mechanistic models and empirical models. Mechanistic models are complex in structure and require many localized parameters [7]. Empirical models are simple with fewer parameters, and are applicable on a wide scale [8]. In 1996, Howarth et al. [9] first used the indicator of net anthropogenic nitrogen input (NANI) to assess the impact of human activities on regional nitrogen balances. This method uses data on the main nitrogen inputs such as net nitrogen input from food/feed, nitrogen fertilizer consumption, crop nitrogen fixation, and atmospheric nitrogen deposition to estimate the intensity of nitrogen input due to human activities in regions. The method can effectively circumvent the complexity and stochasticity of nitrogen transport processes, and has been effectively validated in regional nitrogen pollution assessment [10–13] and nitrogen balance studies [14–16].

The variability in scale and parameter selection significantly impacts on the estimation of NANI and identification of nitrogen emission sources. Large-scale studies, such as global, regional and national scales, provide a macroscopic view to understand the spatial and temporal variabilities of NANI, but have a relatively low data precision. Small-scale studies provide both a higher data precision and have an advantage of being reliable in identifying sources of nitrogen pollution [17–19]. At the same time, the factors influencing NANI are diverse and regionally variable. In agricultural production areas, such as the Lake Michigan Basin in USA [10] and the Huaihe River Basin in China [20], the main source of N input is fertilizer input. In contrast, in the highly urbanized Shenzhen region of China, the main source of nitrogen input is food nitrogen [21]. In the Dianchi [22] and Taihu basins [23] in China, N inputs are positively correlated with population density and arable land area and negatively correlated with forest and grassland area, but in the Zhanjiang Bay region of China, the NANI were mainly associated with diet structure changes [24]. In fact, although the factors influencing NANI are diverse, changes in these factors are usually associated with the Intra- restructuring within agricultural industry or with its transition to secondary and tertiary industries. The negative environmental effects triggered by the transition from primary to secondary industries are often more severe than those of primary industries, especially in developing countries with inefficient environmental management [25]. Therefore, understanding the impact of agricultural structure adjustment on regional NANI allows for the identification of regional N pollution sources from a more macro and global perspective, and the development of target-specific N reduction policies.

Agricultural restructuring is an important perspective for understanding social systems and economic transformation [26]. Dong et al. [27] combined the concept of energy value with a multi-objective linear programming approach to evaluate energy value indicators before and after agricultural structure optimization in Manas County, China. Yu et al. [28] studied the change in agricultural cropping structure in China based on a water footprint and a multi-objective optimization model. Happe [29] studied the changes in agricultural cropping structure in Germany based on agent-based spatial and dynamic simulation model AgriPoliS. In addition, many previous studies based on the roles of

certain factors (e.g., population migration [30], rural decline [31], and fertilizer application to different crops [32]) on agricultural restructuring, but studies on the impact of agricultural restructuring on the environment (e.g., nitrogen and phosphorus pollution) are still deficient. Therefore, a simple model characterizing agricultural restructuring is needed to allow us to examine the interactions between socio-economic and environmental systems and land use issues.

As the world's largest developing country, China has experienced a dramatic change in its agricultural industry structure over the past 30 years. The Pearl River Basin is one of the seven major river basins in China, encompassing three different regional economic characteristics in eastern, central and western China, and is a typical representative of the dramatic changes in China's agricultural industrial structure with marked regional differences. A large number of agricultural industries have shifted to secondary and tertiary industries from 1990 to 2019 in the Pearl River Basin. In the last 30 years, the proportions of agricultural industries in the downstream of the basin decreased by 78.5% and 76.9% in Fujian and Guangdong provinces, respectively. In contrast, ones in the three upstream provinces of Guangxi, Guizhou and Yunnan fell by an average of 63.4%. In the process of agricultural structure adjustment, arable land areas have been decreasing, and the amount of pesticide and fertilizer inputs per unit area have been increasing, which promote the growth of regional GDP but exacerbate environmental pollutions in the Pearl River Basin. In this paper, the Pearl River Basin was taken as an example to explore the impact of agricultural structure adjustment on the NANI over the past 30 years. The main objectives of this study include: (1) to study the spatial and temporal patterns of NANI at the municipal scale in the Pearl River Basin; (2) to analyze the structural changes of nitrogen input sources in the Pearl River Basin; (3) to quantify the temporally varying characteristics of the impact of agricultural factors on the NANI in the Pearl River Basin. The results not only help to reveal the mechanism of regional variability of NANI, but also facilitate the development of targeted nitrogen pollution control policies in the Pearl River Basin.

## 2. Materials and Methods

### 2.1. Overview of the Study Area

The Pearl River basin covers 49 cities in five provinces, namely Yunnan, Guizhou, Guangxi, Guangdong and Fujian (Figure 1), with a basin area of $44.21 \times 10^4$ km$^2$. Under the influence of the subtropical monsoon, the Pearl River basin has a mild and rainy climate, with an average annual temperature of 14–22 °C and an average annual rainfall of 1200–2200 mm. The agricultural arable land accounts for 18.3% of the entire basin area, and the main crops include rice, corn, sugar cane and citrus, etc. The main types of industry are sugar, food, chemical, metallurgy, etc. There are large regional differences in economic development. The upstream cities have slow economic development due to poor natural conditions , while the downstream cities in Guangdong and Fujian provinces have developed economies. In 2015, the average urbanization rate reached 53% and its GDP accounted for 17% of China's total GDP in the Pearl River Basin. In particular, the urbanization rates in Guangdong and Fujian provinces were 74.15% and 68.8% in 2020, respectively.

### 2.2. Data Acquisition

The data used in this study are annual statistics for 1990–2019 for all 49 cities within the Pearl River Basin. The sources of these statistics include: the China Urban Statistical Yearbook (1990–2019), the China Environmental Yearbook (1990–2019), the 1990–2019 Agricultural Statistical Yearbook of the selected cities, and relevant statistical information such as the National Economic and Social Development Statistical Bulletin. The NANI values for all cities in the basin were estimated over the past 30 years, by using agricultural, industrial, population, precipitation and environmental statistics.

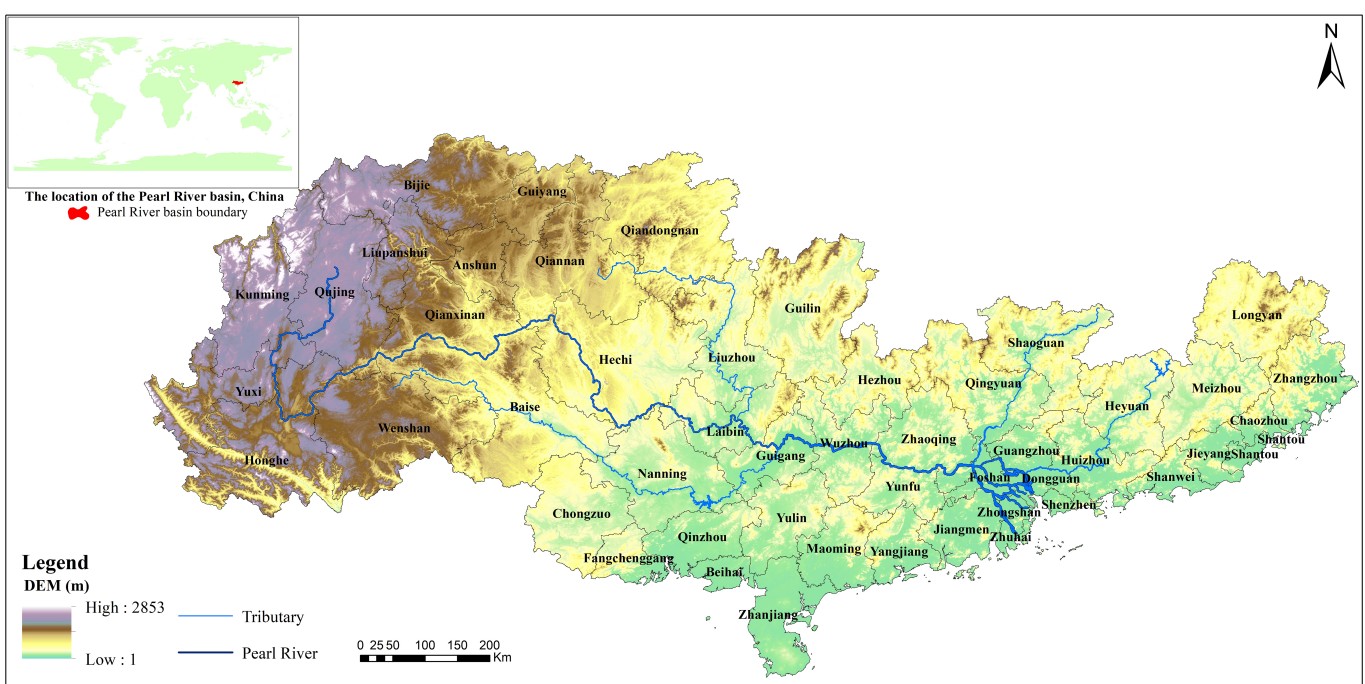

**Figure 1.** The location of the Pearl River Basin and the spatial distribution of its municipal cities.

*2.3. Analysis Methods*

2.3.1. Estimation of NANI Values

The nitrogen input sources of NANI include food/feed nitrogen, fertilizer nitrogen, crop nitrogen fixation, and atmospheric nitrogen deposition. This study does not consider the nitrogen that enters the redistribution and recycling process such as sewage discharge, human feces, etc., as part of NANI. The NANI values are calculated in the following Equations (1) and (2) [9]. In Equation (1), $N_{im}$ is food/feed nitrogen input, $N_{fer}$ for fertilizer nitrogen input, $N_{cro}$ for crop nitrogen fixation, and $N_{dep}$ for atmospheric nitrogen deposition, respectively. The $N_{im}$ is calculated in Equation (2).

$$NANI = N_{im} + N_{fer} + N_{cro} + N_{dep} \tag{1}$$

$$N_{im} = N_{hf} + N_{lfnc} - N_{lpn} - N_{cpn} \tag{2}$$

In Equation (2), $N_{hf}$, $N_{lfnc}$, $N_{lpn}$ and $N_{cpn}$ stand for human food nitrogen consumption, livestock feed nitrogen consumption, livestock production nitrogen, and crop production nitrogen, respectively. They are estimated by the equations of (3), (4), (5) and (6).

$$N_{hf} = \frac{(P_u \times D_u + P_r \times D_r) \times 365}{6.25 \times 10^6} \tag{3}$$

In Equation (3), $P_u$, $P_r$, $D_u$ and $D_r$ are the number of urban population, the number of rural population, the average daily protein intakes (g person$^{-1}$) of urban and rural population, respectively.

$$N_{lfnc} = \sum_{i=1}^{m} (LSN_i \times LSDN_i \times 10^{-3}) \tag{4}$$

$$N_{lpn} = \sum_{i=1}^{n} \left[ LSN_i \times (LSDN_i - LSEN_i) \times K_S \times 10^{-3} \right] \tag{5}$$

In equations of (4) and (5), $m$, $LSN_i$, $LSDN_i$, $LSEN_i$ and $K_s$ are the number of livestock species, the number of livestock, the amount of nitrogen required for livestock growth (kg capita$^{-1}$) and the amount of nitrogen excreted by livestock (kg capita$^{-1}$), and the edible fraction of livestock (%), respectively.

$$N_{cpn} = \sum_{i=1}^{n}(CP_i \times LF_i) \tag{6}$$

In Equation (6), $n$, $CP_i$ and $LF_i$ are the number of crop species, crop yield of the ith crop and the nitrogen fixation rate (%) of the ith crop, respectively. The selection of relevant parameters for the estimation of NANI values was described below.

**Nitrogen fertilizer application** ($N_{fer}$): the $N_{fer}$ is the sum of the amount of pure nitrogen in the nitrogen fertilizer and 20% of compound fertilizer.

**Crop nitrogen fixation** ($N_{cro}$): considering the small number of crop species for which N fixation coefficients are currently available, only the estimates for three crops of soybean, peanut and rice in the study area were counted in this paper. The nitrogen fixation coefficients for soybean, peanut and rice, were 9600 kg km$^{-2}$ y$^{-1}$, 8000 kg km$^{-2}$ y$^{-1}$, and 4480 kg km$^{-2}$ y$^{-1}$ [14,33], respectively.

**Atmospheric nitrogen deposition** ($N_{dep}$): the $N_{dep}$ consists of two components of wet and dry deposition. Wet deposition of nitrogen is the product of the annual regional rainfall and nitrogen concentrations in the rainfall (the average nitrogen concentration in precipitation of each city is 1.98 mg L$^{-1}$ [34]). Based on a 3:7 ratio of wet to dry atmospheric nitrogen deposition [22], the dry deposition of nitrogen in the study area was derived from the wet deposition of nitrogen.

**Human food nitrogen consumption** ($N_{hf}$): in this paper, the daily N consumption per capita in urban and rural areas was calculated at 69 g and 64.6 g, respectively [15].

**Nitrogen consumption in livestock/aquaculture feeds** ($N_{lfnc}$): nitrogen intakes during farming of pigs, cattle, sheep, poultry and aquatic products were mainly calculated. The annual nitrogen intake coefficients for relevant livestock and aquatic products were categorized as 54.82 kg y$^{-1}$ for cattle, 9.5 kg y$^{-1}$ for pigs, 14.45 kg y$^{-1}$ for sheep, 0.188 kg y$^{-1}$ for poultry and 29.4 g kg$^{-1}$ for aquatic products [15,17,35].

**Crop/fruit N production** ($N_{cpn}$): the $N_{cpn}$ is the sum of the N yields of different crops/fruits. There are 12 major crops /fruits in the study area and their N contents were taken as 5.62% for soybean, 1.93% for peanut, 1.18% for rice, 0.32% for vegetables, 1.79% for wheat, 1.4% for maize, 0.32% for potato, 4.46% for rapeseed, 0.19% for sugarcane, 0.17% for citrus, 0.93% for banana and 0.32% for pear [16,35,36].

### 2.3.2. Nitrogen Input Source Index of NANI

The nitrogen input sources of NANI can be divided into agricultural and non-agricultural sources. The main agricultural N inputs are fertilizer N, crop N fixation, livestock feed N and atmospheric N deposition from agricultural sources, while the non-agricultural N are atmospheric deposition of N from non-agricultural sources and human food N. NANI changes are closely related to industrial and agricultural development [37,38]. The transition from agriculture to secondary and tertiary sectors inevitably leads to a reduction in arable land area, population concentration and changes in traditional cultivation and farming, which finally affect the structure of the NANI input sources. For example, in Shenzhen City with a highly urbanized and minimally agricultural areas, the NANI is dominated by food N inputs from non-agricultural sources [21]. Therefore, we proposed a novel nitrogen input source index of NANI, namely the ratio of agricultural nitrogen inputs to non-agricultural nitrogen inputs of NANI (ASNA), to indirectly reflect the effect of agricultural structure adjustment on NANI. Intuitively, ASNA > 1 indicates a large contribution of nitrogen inputs from agricultural sources, while ASNA < 1 indicates a large nitrogen inputs from non-agricultural sources. The ASNA is calculated according to the Equation (7).

$$ASNA = \frac{N_{fer} + N_{cro} + N_{lfnc} - N_{lpn} - N_{cpn} + N_{Adep}}{N_{hf} + N_{Idep}} \tag{7}$$

### 2.3.3. Spatial Analysis and Statistics

Arcgis 10.3 software was used to output spatial distribution maps of NANI and ASNA in the study area. The hotspot analysis module of Arcgis 10.3 software was also used to

identify hotspots of ASNA over the last 30 years. The contributions of six agricultural structure indicators (non-agriculture GDP, agriculture GDP, agricultural land area, population density, nitrogen fertilizer consumption and Livestock farming density) to the ASNA variance in the study area was estimated by using the R package (rdacca.hp).

## 3. Results

### 3.1. Spatio-Temporal Characteristics of NANI at the Watershed and Municipal Scales

At the basin scale, the annual mean of NANI in the Pearl River basin increased from 15,683 kg N $km^{-2}$ $y^{-1}$ in 1990 to 19,461 kg N $km^{-2}$ $y^{-1}$ in 2015, and then decreased to 16,178 kg N $km^{-2}$ $y^{-1}$ in 2019 (Figure 2). Overall, NANI in the Pearl River Basin was significantly higher than the average value of 5013 kg N $km^{-2}$ $y^{-1}$ over China [39].

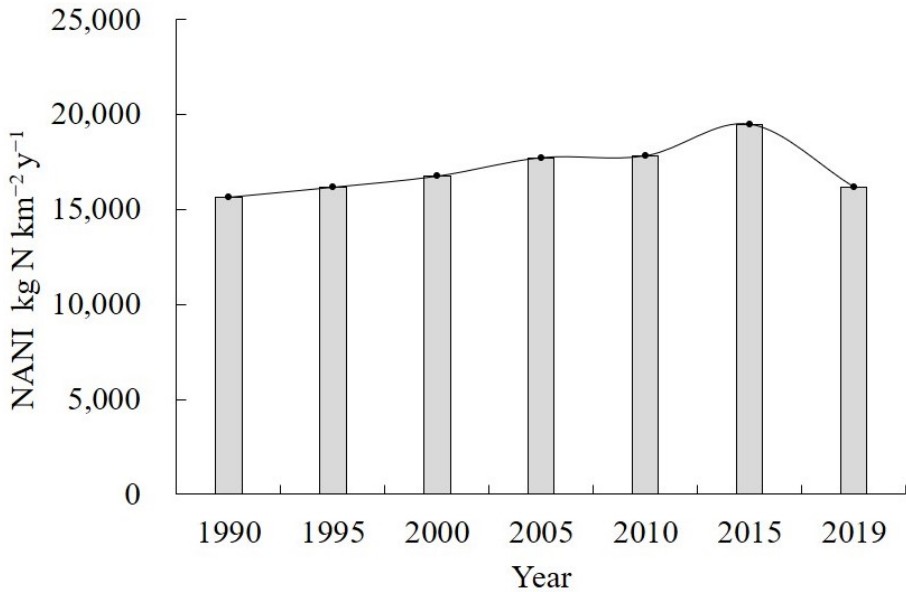

**Figure 2.** Interannual variation of NANI in the Pearl River Basin from 1990 to 2019.

At the municipal scale, from 1990 to 2019, 75.5% of the 49 cities in the Pearl River Basin also showed a trend of increasing and then decreasing NANI values, with the peak occurring around 2015 (Figure 3). A total of 18.4% of cities were in a continuous decline in NANI, while 6.1% of cities had a continuous upward trend. For example, Guangzhou City decreased from 31,497 kg N $km^{-2}$ $y^{-1}$ in 1990 to 14,643 kg N $km^{-2}$ $y^{-1}$ in 2019, with a decrease of 58.3%, and Shenzhen City increased from 10,454 kg N $km^{-2}$ $y^{-1}$ in 1990 to 33,894 kg N $km^{-2}$ $y^{-1}$ in 2019, with an increase of 320%. In terms of the spatial distribution of NANI, the economically developed cities in the downstream of the basin consistently have higher NANI than the relatively economically backward cities in the upstream. The mean values of NANI in downstream coastal cities such as Guangdong, Shenzhen, Shantou, Jieyang, Beihai and Zhangzhou reached 23,267 kg N $km^{-2}$ $y^{-1}$ in the last 30 years, which were two to three times higher than that of economically less developed upstream cities such as Kunming, Bijie, Wuzhou and Yuxi (mean value of 7965 kg N $km^{-2}$ $y^{-1}$). Overall, the estimated NANI at both watershed and municipal scales demonstrated a trend of increasing NANI before 2015 and decreasing NANI after 2015. This may be related to the implementation of a nationwide agricultural fertilizer application reduction policy in China starting in 2015.

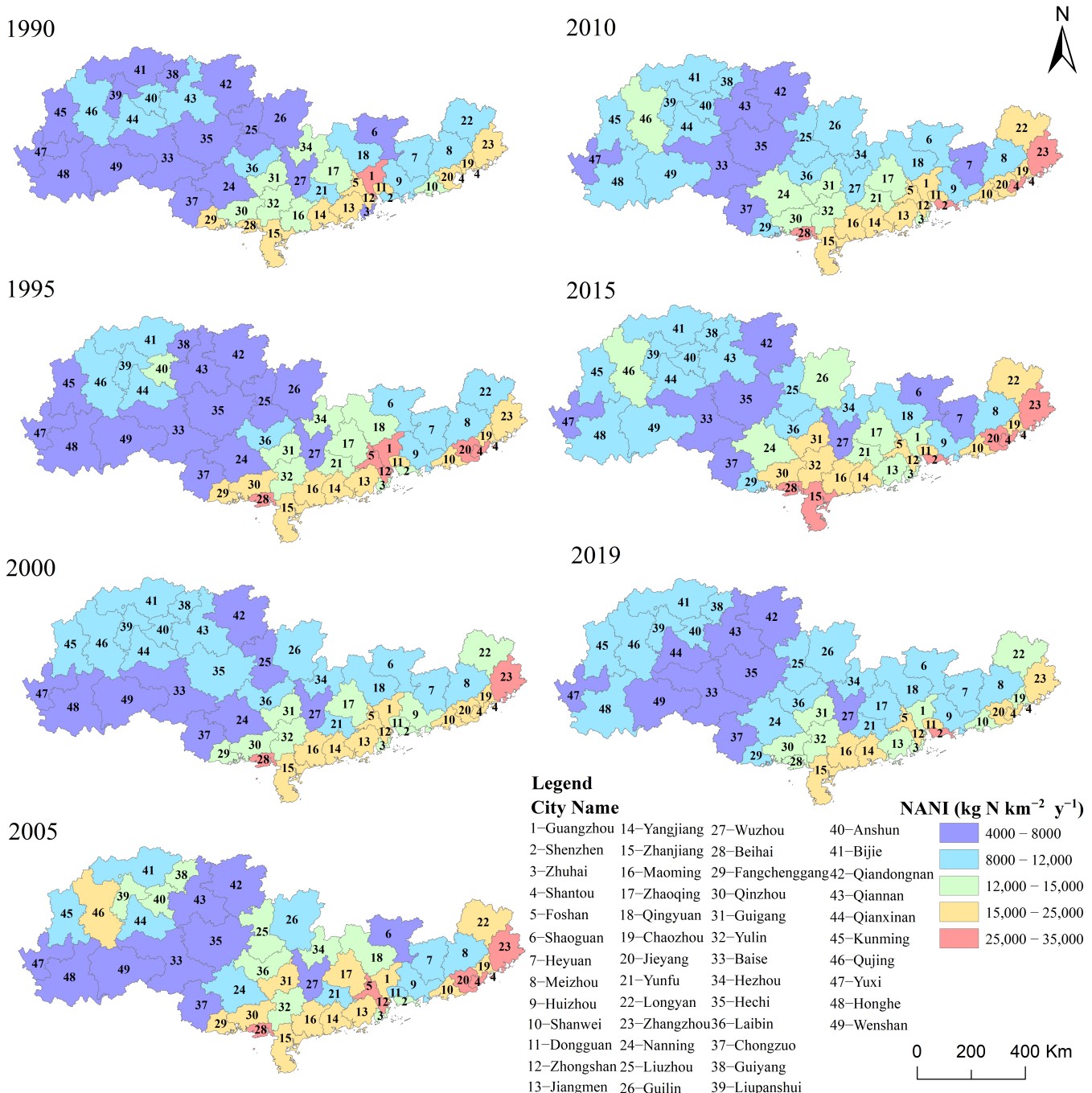

**Figure 3.** Spatial and temporal distributions of NANI values at the municipal scale in the Pearl River Basin from 1990 to 2019.

### 3.2. Variation in NANI's Nitrogen Input Sources Across the Basin

From 1990 to 2019, there was a clearly spatial heterogeneity of NANI input sources in the Pearl River Basin (Figure 4). The NANI in economically developed downstream areas was dominated by food/feed N (49.6% of total N inputs on average), while fertilizer N in upstream areas with relatively poor economic development accounted for 54.9% of N inputs on average. In contrast, in economically developed downstream cities, fertilizer N and crop N decreased significantly, and the share of food/feed input N and atmospheric deposition N increased significantly. For example, fertilizer N inputs in Guangzhou decreased from 70.9% in 1990 to 29.3% in 2019, and food/feed input nitrogen increased from 11.4% to 24.3%. The food/feed N inputs in Shenzhen increased twofold and the atmospheric deposition

of nitrogen decreased to 33.3% over the past 30 years. However, a few downstream cities such as Zhangzhou, Heyuan, Meizhou and other coastal cities had an increase in fertilizer N of up to 50% due to the emphasis on agricultural development. Regional differences in nitrogen inputs were more related to differences in agricultural industry structures among the upstream and downstream areas. Due to the slow restructuring of agriculture, the middle and upstream cities had a larger share of agricultural industries than the downstream cities. Downstream cities had high urbanization rates, decreasing arable land, high population density and a smaller share of agricultural industries.

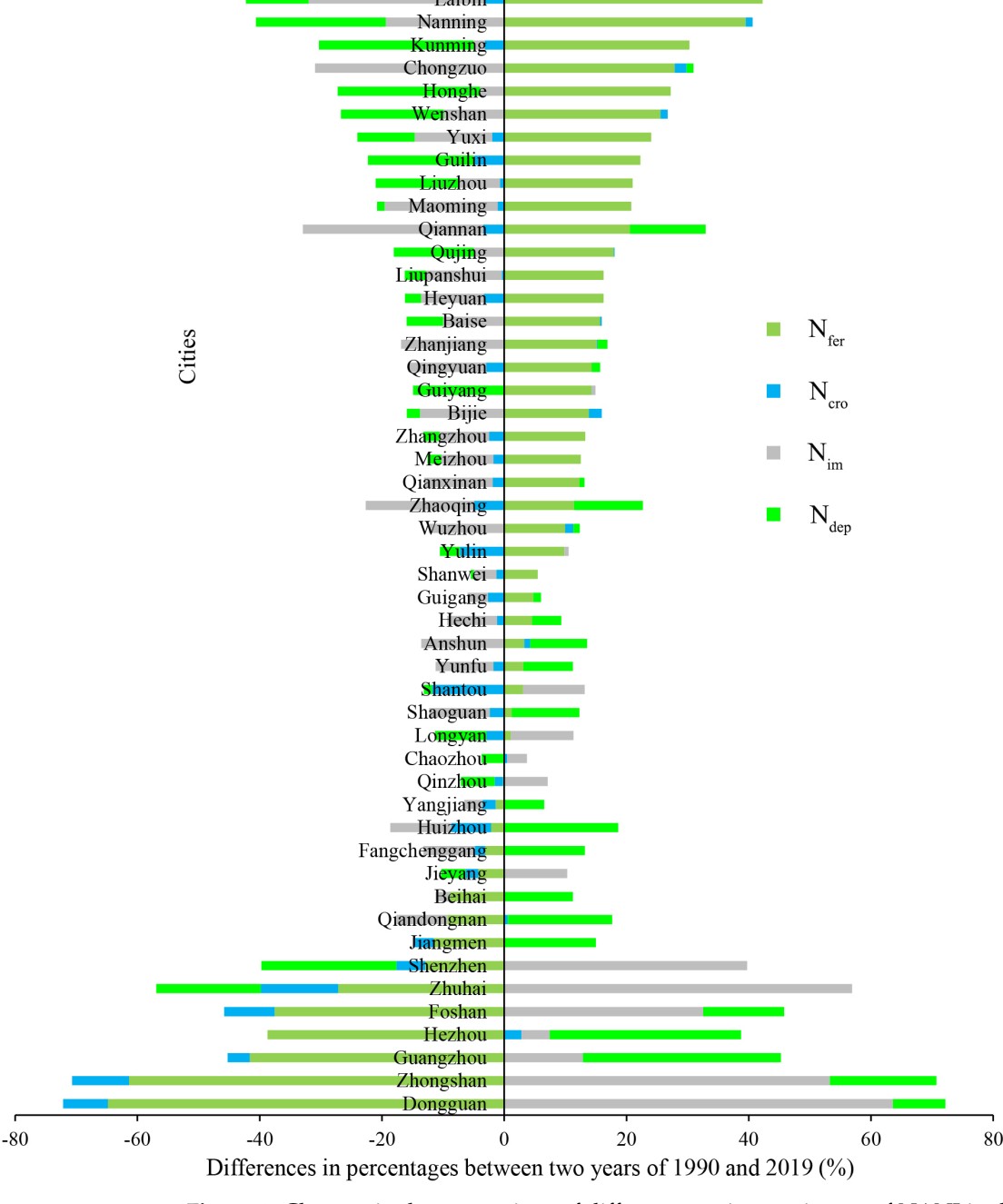

**Figure 4.** Changes in the proportions of different net nitrogen inputs of NANI in the Pearl River Basin cities between the two years of 1990 and 2019.

### 3.3. Spatio-Temporal Patterns of the ASNA Values from 1990 to 2019

The ASNA values in the Pearl River Basin showed a trend of increasing and then decreasing over the past 30 years (Figure 5). It increased from 1.19 in 1990 to 1.36 in 2005 and

then decreased to 0.87 in 2019. This trend coincides with the NANI changes in the study area. At the municipal scale, upstream cities demonstrated an increasing trend of ASNA values (Figure 6). For example, the ASNA values of Nanning and Kunming increased by 86% and 130%, respectively, from 1990 to 2019. Downstream cities demonstrated a decreasing trend in ASNA values. For example, the ASNA value decreased from 0.9 in 1990 to 0.1 in 2019, with a decrease of 89% in Shenzhen. However, in some midstream and downstream cities, the changes of ASNA were relatively exceptional. For example, downstream coastal cities such as Longyan, Zhangzhou, Beihai, Zhanjiang and Maoming had high ASNA values (ASNA > 1), midstream cities such as Qiandongnan, Guilin, Liuzhou, Hechi and Baise had low ASNA values all years round (ASNA < 1).

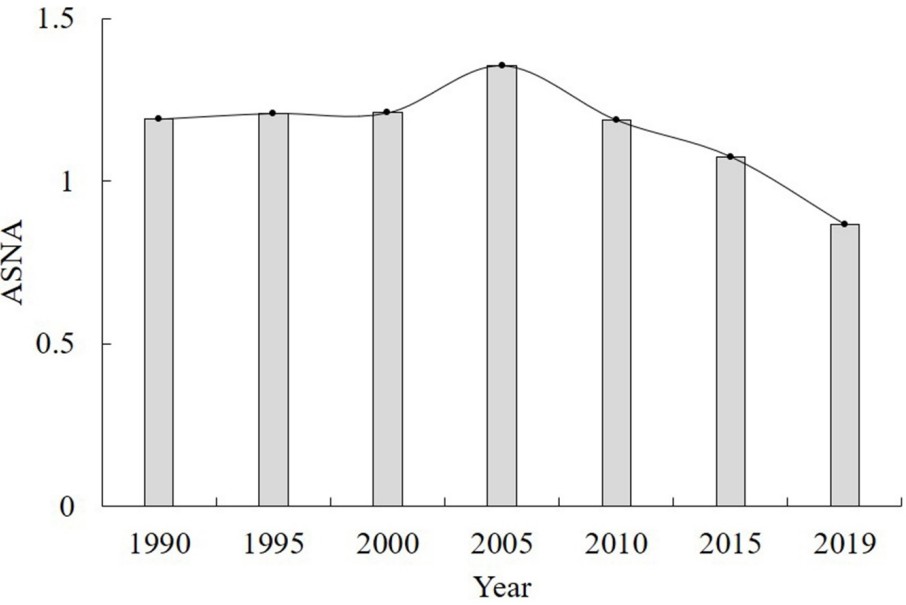

**Figure 5.** Interannual variation in ASNA values in the Pearl River Basin from 1990 to 2019.

Hotspots of ASNA in the study area was Further identified by Getis-Ord Gi* method. The results demonstrated that the distribution of ASNA hotspots changed significantly over the past 30 years (Figure 7). In 1990, the ASNA hotspots were mainly Guangzhou, Dongguan, Huizhou, Zhongshan, Jiangmen, Yulin, Beihai and Zhanjiang (ASNA values ranged from 1 to 4), while the cold spot of ASNA distribution was Liuzhou (ASNA of 0.4). By 2019, two hotspots of ASNA distribution occurred (confidence level >90%), namely the upstream cities centered on Kunming and the downstream cities centered on Zhanjiang and Yulin. Moreover, the hotspots (Shenzhen, Guangzhou, Dongguan, Huizhou, Zhongshan and Jiangmen, etc.) in 1990 turned out to be cold spots in 2019.

### 3.4. Quantifying the Influence of Agricultural Factors on ASNA

The above findings indicated that the spatial and temporal changes in NANI were influenced by agricultural industry restructuring in the study area. To reveal the influence of agricultural industry restructuring on NANI, Linear regression models were used to analyze the association between ASNA and agricultural factors associated with agricultural industry for the two periods of 1990 and 2019 in the study area. First, we clustered ASNA data for two years, between 1990 and 2019, to form five urban clusters with ASNA values of 0–0.5, 0.5–1, 1–1.5, 1.5–2.5, and 2.5–3.5, respectively. Then, the correlation between indicators of agricultural industry factors (Agriculture GDP, Non agriculture GDP, Agricultural land area, Population density, Nitrogen fertilizer application and Livestock farming density), and ASNA in urban clusters was analyzed using linear regression models (Figure 8). Except for the factor of the agriculture GDP and non-agriculture GDP, which were not significantly correlated with ASNA ($p > 0.05$), the remaining four agricultural factors were significantly

correlated with ASNA in 1990 ($p < 0.05$). By 2019, ASNA only had a significant correlation with the agricultural land area ($p < 0.05$) and insignificant correlation with other factors ($p > 0.05$), especially when negatively correlated with two indicators of non-agriculture GDP and population density. The contributions of traditional farming and livestock and aquaculture to NANI decreased in the Pearl River Basin as the agricultural industry shifted to secondary and tertiary industries.

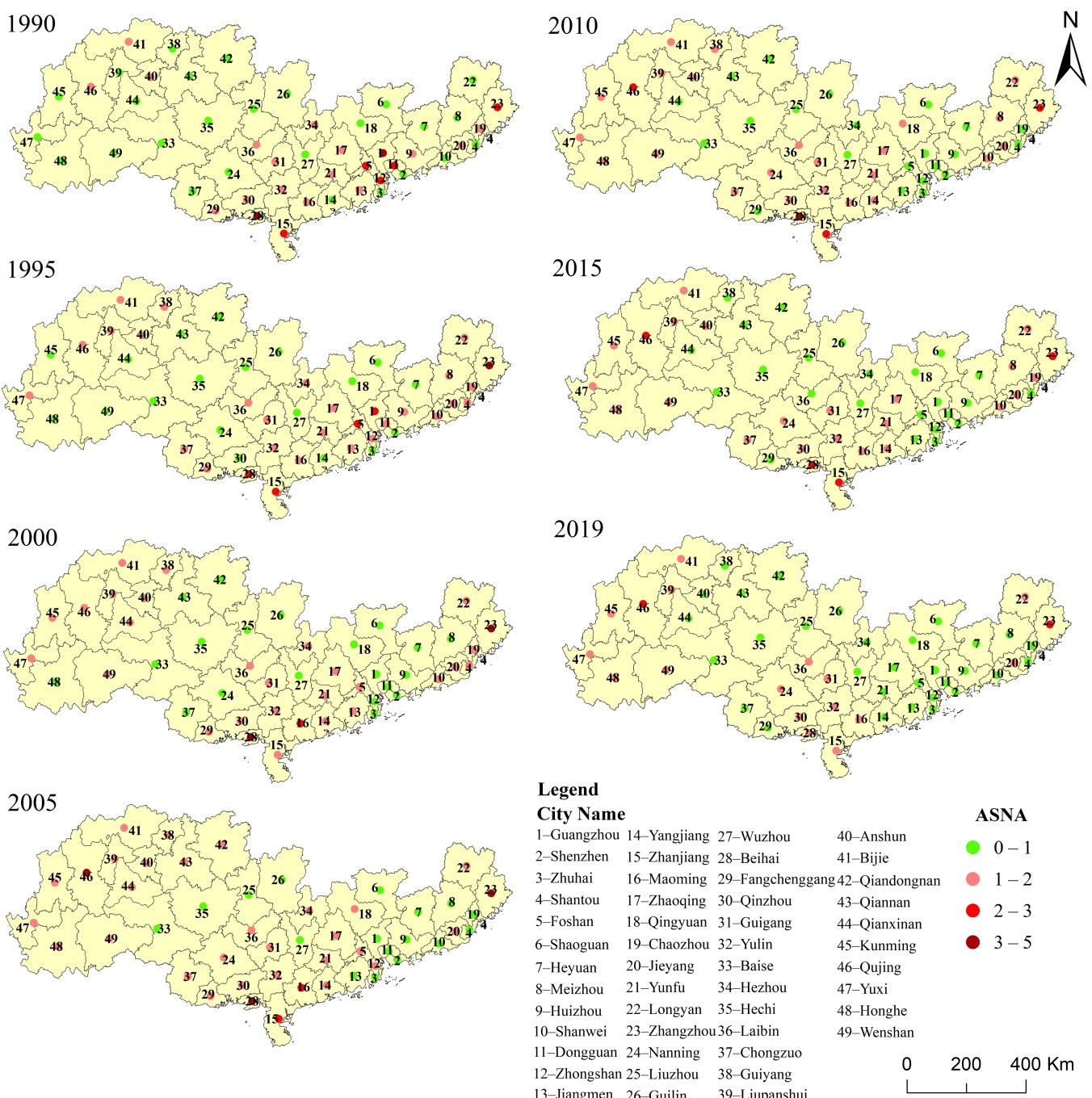

**Figure 6.** Spatial and temporal distribution patterns of ASNA in the Pearl River Basin from 1990 to 2019.

The contributions of the above six agricultural factors to the variance of ASNA were further estimated using the hierarchical variance decomposition method. The estimation presented that the influence of the agricultural factors on ASNA occurred with a significant decreasing trend over the last 30 years. In 1990, the total contribution of six agricultural

factors to the variance of the ASNA amounted to 81.5%, whereas by 2019 it fell to 20.4%, with a decrease of 75% (Figure 9). The contribution of the different agricultural factors to the variance of ASNA varied significantly. In 1990, the magnitude of explanation of the variance of ASNA variance by the six agricultural factors was in the following order: Nitrogen fertilizer consumption (38.3%) > agricultural land area (25.3%) > livestock farming density ( 13.7%) > population density (5.1%) > non-agriculture GDP (−0.47%) and agriculture GDP (−0.47%). In contrast, by 2019, the ranking of the magnitude of the agricultural factor in explaining the variance of ASNA was: agricultural land area (7.1%) > population density (5.7%) > livestock farming density (3.1%) > non-agriculture GDP (1.9%) and agriculture GDP (1.8%) > nitrogen fertilizer consumption (0.8%). Overall, the influence of cultivation and farming on ASNA in the Pearl River Basin demonstrated a decreasing trend, while the influence of non-agricultural factors had an increasing trend. This indicated that although agricultural structure adjustment in the Pearl River Basin led to a significant increase in the N inputs of non-agricultural sources, agricultural production activities associated with agricultural land use continued to be one of the main N sources of NANI.

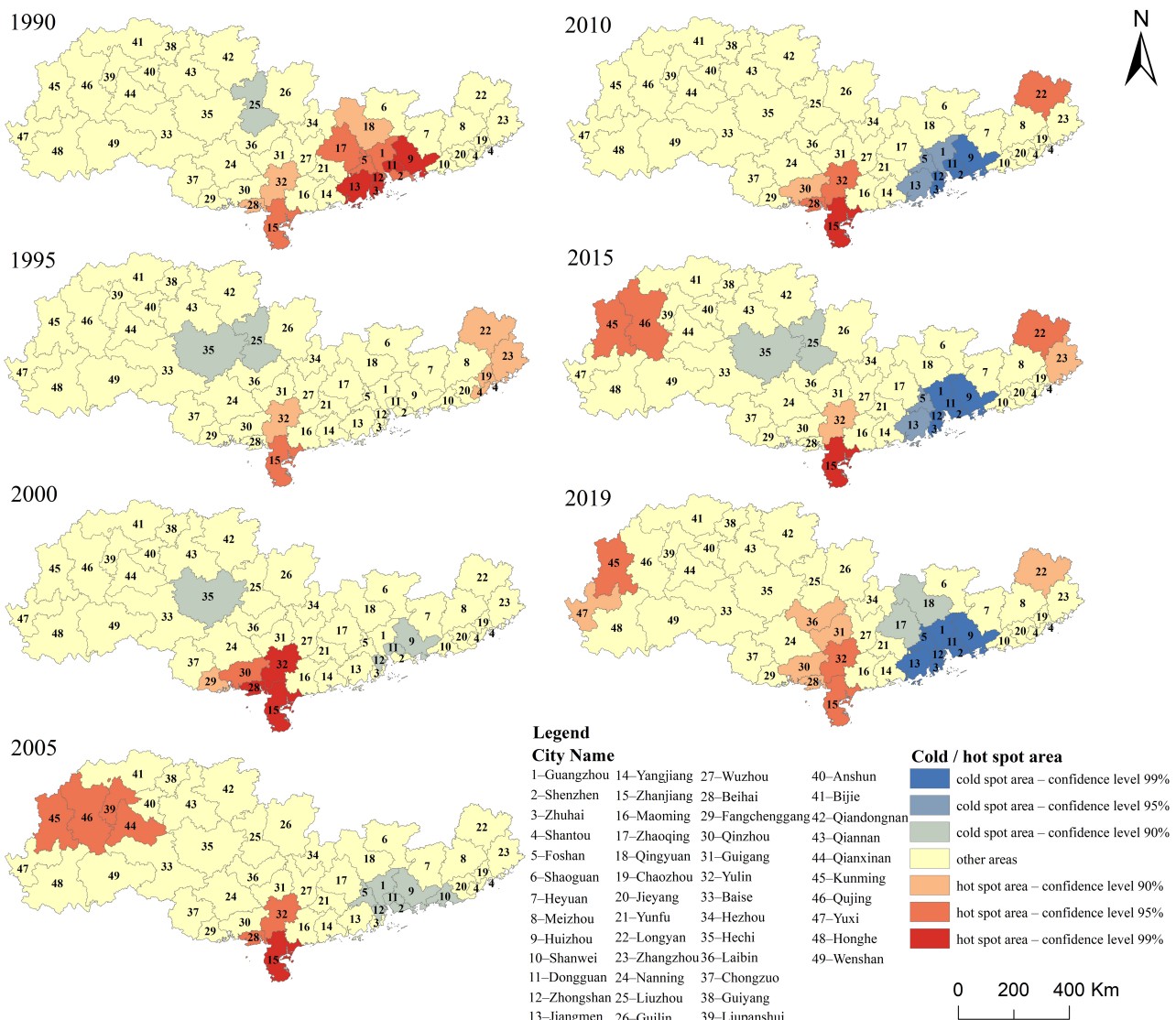

**Figure 7.** Cold/hot spots in the ASNA distribution in the Pearl River Basin from 1990 to 2019.

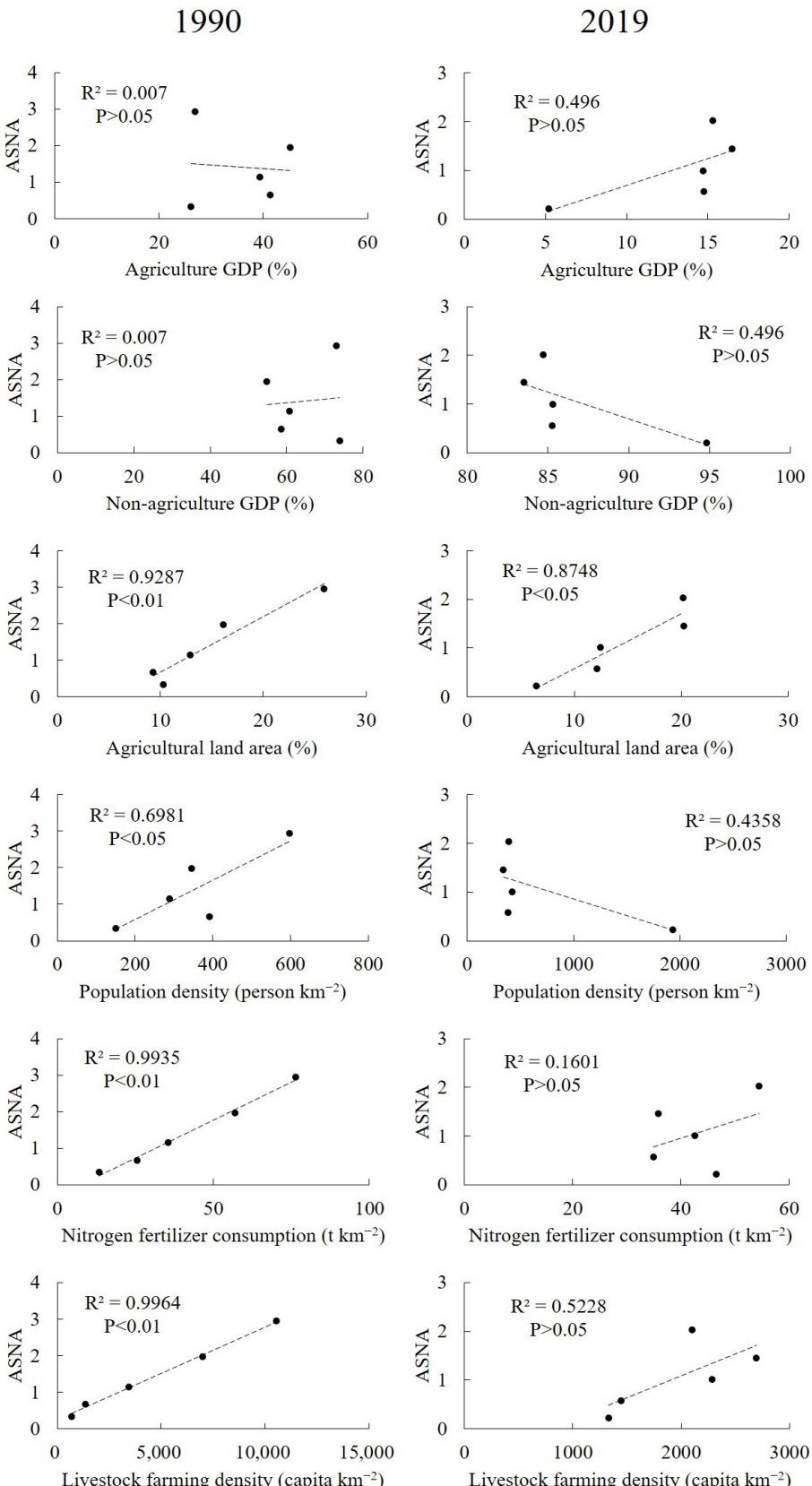

**Figure 8.** Correlation of ASNA with six agricultural factors in the years of 1990 and 2019.

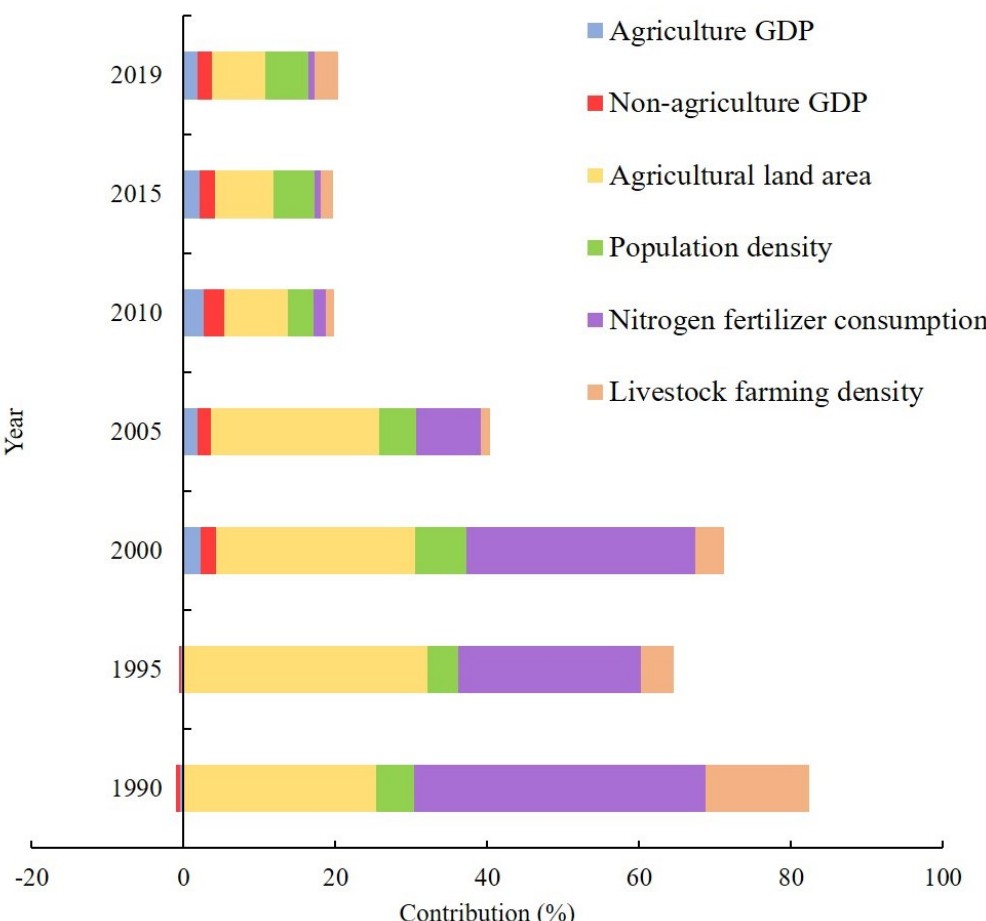

**Figure 9.** Changes in the contributions of agricultural factors to the ASNA variances from 1990 to 2019.

## 4. Discussion

### 4.1. Comparison of NANI and Its N Input Sources with Other Regions Worldwide

The mean NANI values of 17,108 kg N km$^{-2}$ y$^{-1}$ in the Pearl River Basin is higher than the global average (1044 kg N km$^{-2}$ y$^{-1}$) [40]. Compared to other basins in the world, the NANI values in the Pearl River basin was also higher than 5861 kg N km$^{-2}$ y$^{-1}$ in the Michigan basin in the USA [10] and 8800 kg N km$^{-2}$ y$^{-1}$ in the Baltic Sea basin in Europe [11]. Compared to other basins in China, NANI values in the Pearl River basin were higher than those in the Yellow River basin (6787 kg N km$^{-2}$ y$^{-1}$) [35], the Yangtze River basin (7297 kg N km$^{-2}$ y$^{-1}$) [41], and the Three Gorges Reservoir Area (12,399 kg N km$^{-2}$ y$^{-1}$) [16], and lower than that in the Huaihe River basin (28,000 kg N km$^{-2}$ y$^{-1}$) [20].

In terms of N input sources of NANI, fertilizer N, atmospheric N deposition and food/feed N inputs were dominant in the Pearl River Basin (Figure 4), which was similar to the results of some existing studies [15,16,35,42]. However, fertilizer N in the Pearl River Basin tended to decrease but remained large, while food/feed input N and atmospheric deposition N tended to increase. This is related to agricultural industrial restructuring and geoclimatic conditions in the Pearl River Basin. The reduction in the total arable land areas and the increase in nitrogen fertilizer inputs per unit area of agricultural production (annual average input of 6449 kg km$^{-2}$ y$^{-1}$) led to a larger share of fertilizer nitrogen in the Pearl River Basin (40% in 2015, 41% in 2019) despite the nationwide fertilizer reduction policy starting in 2015. In particular, the rapid development of secondary and tertiary industries in the downstream areas brought an extremely high urban population density and a massive increase in demand for meat, eggs and milk. This has naturally triggered an

increase in food/feed input nitrogen in the basin. In addition, there is a large proportion of forested land (about 82%) and abundant rainfall in the study area. In the Pearl River Basin, the rapid development of secondary and tertiary industries has caused large quantities of nitrogen pollutants into the atmospheric system over the last 30 years [43]. Therefore, atmospheric nitrogen pollution enters the surface ecosystem as a result of atmospheric and water circulation, which leads to an increase in atmospheric nitrogen deposition in the study area.

There were regional differences in the N input sources of NANI in the Pearl River Basin. Similar to the Michigan Basin in the USA [10] and the Huaihe River Basin in China [20] , the N input sources in the relatively economically underdeveloped areas of the upstream areas were dominated by fertilizer N consumption and atmospheric N deposition with an average percentage of 36% and 41%, respectively, over the last 30 years. Similar to other regions in the world with high urbanization levels [21,44], the nitrogen input sources in the economically developed areas of the downstream in the Pearl River Basin were dominated by food input nitrogen and atmospheric nitrogen deposition, with an average share of 33% and 31%, respectively, over the last 30 years. The regional variability was mainly related with the differences in the agricultural structure among cities in the study area. In general, the upstream areas are economically backward compared to the downstream areas, and the agricultural industry is still relatively dominated by traditional agricultural planting and farming. In contrast, secondary and tertiary industries are dominant in the economically developed downstream areas, and a small proportion of agricultura planting and farming remains. This explains well the fact that the nitrogen fertilizer inputs were dominant in relatively underdeveloped upstream areas and the predominance of food nitrogen inputs were in economically developed downstream areas.

### 4.2. Sensitivity of ASNA Values to Agricultural Restructuring

The changes in NANI are closely related to the regional agricultural development [37,38]. The NANI was calculated directly or indirectly by agricultural industry factors. It is difficult to directly quantify the impact of changes in agricultural industry structure on NANI. First, the covariance between intra- agricultural industry factors is significant. Second, there is a direct or indirect correlation between NANI and agricultural industry factors. For this reason, The ASNA indicators was first proposed to indirectly reflect the impact of agro-industrial restructuring on NANI. Intuitively, ASNA > 1 indicates a large contribution of nitrogen inputs from agricultural sources, while ASNA < 1 indicates a large contribution of nitrogen input from non-agricultural sources.

The results demonstrate that the ASNA indicator could effectively characterize the spatial and temporal variation of NANI in the study area. On the one hand, the changes of the ASNA in the Pearl River Basin over the past 30 years (Figure 5) is consistent with that of NANI (Figure 2), i.e., it shows an increasing trend before 2015 and a decreasing trend after 2015. On the other hand, the results of the hotspot analysis of ASNA also confirmed that the economically developed downstream cities (Guangzhou, Shenzhen, Dongguan and Huizhou, etc.) shifted from hotspots in 1990 to cold spots in 2019 (Figure 7). This matches well with the reality of the changes of agricultural industry structure in those cities. At the early stage of urbanization in the Pearl River Basin, the share of planting and farming in the downstream cities of Guangzhou, Shenzhen, Dongguan and Huizhou was significantly higher than that of the surrounding cities, and thus became the hotspots of the ASNA distribution in 1990. However, after nearly 30 years of urbanization, the agricultural industries in these cities declined significantly while the secondary and tertiary industries developed rapidly, which caused a continuous decrease in the ASNA values of these cities. Finally, these cities became the cold spots of the ASNA distribution in 2019.

The ASNA had a decreasing trend from 1990 to 2019 with urbanization in the Pearl River Basin. This indicates that the contribution of nitrogen input from agricultural sources decreased with comparison with that from non-agricultural sources. The results of the hierarchical variance decomposition also confirmed the decreasing contributions of agricultural

land area, nitrogen fertilizer consumption and Livestock farming density to the ASNA in the Pearl River Basin in the last 30 years (Figure 9). Existing studies also indicate that the contribution of non-agricultural N to the N balance increases with increasing urbanization rates [45–49]. This further suggests that ASNA indicators can effectively characterize spatial and temporal changes in NANI and reveal well the impact of the regional agricultural structure adjustment on NANI.

*4.3. Implications and Recommendations for the Nitrogen Pollution Management in the Pearl River Basin*

The NANI decreased in the Pearl River Basin due to agricultural restructuring, but the N input sources are still dominated by fertilizer N, food/feed N and atmospheric N deposition. There are also significant regional differences in the structure of nitrogen input sources. Therefore, the management of nitrogen pollution in the Pearl River Basin should be based on regional differences in nitrogen input sources. In the relatively economically underdeveloped upstream areas, traditional cultivation and livestock farming continue to contribute a large proportion of nitrogen inputs. The application of inorganic nitrogen fertilizer should be reduced by increasing the return of livestock and poultry manure to the land to reduce inorganic nitrogen input. At the same time, upstream areas such as Yunnan and Guizhou are dominated by karst landscapes, with small areas of usable arable land and large soil fissures. These areas have a low ecological capacity, and soil nitrogen and phosphorus nutrients are prone to contaminating the downstream surface and groundwater through migration or infiltration pollution with surface runoff [50,51]. Therefore, water control in the ecologically fragile upstream areas should be strengthened to reduce soil nitrogen loss. In economically developed downstream areas, nitrogen inputs from traditional cultivation and livestock farming are greatly reduced and nitrogen inputs are dominated by food/feed input nitrogen. Nitrogen inputs from agriculture are no longer a key contributor to highly urbanized environmental problems [52]. Therefore, the focus of prevention and control of nitrogen pollution should be to enhance the denitrification of urban nitrogen emissions and to develop a reasonable and healthy diet. Meanwhile,considering the generally high atmospheric nitrogen deposition in the study area, the optimization and upgrading of industrial structures and production technologies in the watershed should be strengthened to reduce the emission of atmospheric nitrogen pollutants [53].

## 5. Conclusions

The spatio-temporal patterns of NANI in the Pearl River Basin and its driving factors were analyzed during the recent 30-year period from 1990 to 2019. Overall, the NANI decreased in the Pearl River Basin from 1990 to 2019, and was mainly influenced by agricultural structure adjustment. Our proposed ASNA indicator is very effective and sensitive to characterize the effect of agricultural structure adjustment on the NANI in the Pearl River Basin over time. In the future, finer statistical and parameter data at spatial and temporal scales should be used to further reduce the uncertainty in NANI estimates in the study area. Attempts will be also made to verify the validity of the ASNA indicator at different spatial and temporal scales.

**Author Contributions:** Conceptualization, K.X. and J.Z.; Data curation, K.X.; Formal analysis, K.X. and J.Z.; Methodology, J.Z., K.X. and W.W.; Writing—original draft, J.Z. and K.X.; Writing—review and editing, J.Z., K.X., Q.L., W.W. and G.M. All authors have read and agreed to the published version of the manuscript.

**Funding:** This study was supported by the National Natural Science Foundation of China (grant number 41877009 and U20A20114), the Humanities and Social Sciences Foundation of the Chinese Ministry of Education (grant number 18YJA790061), the Social Science Foundation of Jiangsu Province (grant number 18EYB008) and the Natural Science Foundation of Huai'an (grant number HABL202105).

**Institutional Review Board Statement:** Not applicable.

**Informed Consent Statement:** Not applicable.

**Data Availability Statement:** Not applicable.

**Acknowledgments:** We would like to Ying Li and Tianpeng Zhang for their help during the statistical data collection.

**Conflicts of Interest:** The authors declare no conflict of interest.

## Abbreviations

The following abbreviations are used in this manuscript:

| | |
|---|---|
| NANI | Net anthropogenic nitrogen input |
| ASNA | Ratio of agricultural sources to non-agricultural sources in NANI |

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
