# Peer review of "Effect of Agricultural Structure Adjustment on Spatio-Temporal Patterns of Net Anthropogenic Nitrogen Inputs in the Pearl River Basin from 1990 to 2019"

_land, doi:10.3390/land12020311_

Round 1

Reviewer 1 Report

This study presented a new index (ASNA) to analyze the effect of agricultural structure adjustment on net anthropogenic nitrogen inputs (NANI) in the Pearl River Basin, which is interesting and practical. The following suggestions are for the authors to further improve the paper:

1. Strengthen the introduction on agricultural restructuring in the section 1.

2. The method. In the calculation of NANI, non-agricultural sources such as sewage discharge and human waste seem not to be included, why?

3. Figure 1. If you say the Pearl River Basin, it should be the hydrological basin, but the map seems not the hydrological boundary, but the municipal boundaries that overlay the Pearl River Basin? If so, the Legend and the Figure title are not correct. Also, to the north of Heyuan and Meizhou cities, there is a tributary beyond your boundary, why?

4. Lines 150-151. What’s the sources of rainfall nitrogen concentration data? Are they from literatures? Are they different for all the 49 cities? Clarify it in the manuscript.

5. In Figure 8, why are 5 clusters generated to do this analysis rather than using all 49 cities? What do these 5 clusters stand for?

6. Section 4.3. Please give some more specific suggestions on nitrogen management based on the findings of this study and delete those unrelated suggestions. For example, lines 391-393, domestic sewage and industrial wastewater are not analyzed in this manuscript, so that this suggestion is not supported by the results here.

Author Response

Dear  reviewers,

     Thank you for your letter and comments concerning our manuscript entitled “Effect of agricultural structure adjustment on spatial-temporal patterns of net anthropogenic nitrogen inputs in the Pearl River Basin from 1990 to 2019” (Manuscript ID: land-2095581). Your comments have proved very helpful with regard to our revisions and improvements of our paper, as well as providing insightful guidance for our research. We have gone through your comments carefully and have made necessary corrections. We have tried our best to improve the manuscript and we have made many changes to the text according to the comments of reviewers. All changes have been highlighted in red in the revised manuscript.

 1、Strengthen the introduction on agricultural restructuring in the section 1.

Answer: Good suggestion. In the revised draft, we have added a number of advances on agricultural restructuring (in Lines of 76-88).

2、The method. In the calculation of NANI, non-agricultural sources such as sewage discharge and human waste seem not to be included, why?

Answer: Good suggestion. In all literature which uses NANI calculations, the source of nitrogen for NANI usually defaults to an external source of nitrogen entering the watershed, while sewage discharge and human waste are not considered as part of NANI. This is mainly because these processes do not bring new nitrogen, but are processes of redistribution and recycling of nitrogen within the watershed. Therefore, nitrogen from sewage discharge, human waste is also not considered as part of NANI in this study.

3、Figure 1. If you say the Pearl River Basin, it should be the hydrological basin, but the map seems not the hydrological boundary, but the municipal boundaries that overlay the Pearl River Basin? If so, the Legend and the Figure title are not correct. Also, to the north of Heyuan and Meizhou cities, there is a tributary beyond your boundary, why?

Answer: Good suggestion. The boundaries of the Pearl River basin are indeed not real hydrological boundaries, but administrative boundaries. In fact, the only official Chinese data now publicly available on the boundaries of the Pearl River Basin are the administrative boundaries. However, the manuscript is concerned with municipalities within the Pearl River Basin and using the true hydrological boundaries of the basin would affect the research results. In addition, we deleted a tributary beyond the boundary in the updated figure 1.

4、Lines 150-151. What’s the sources of rainfall nitrogen concentration data? Are they from literatures? Are they different for all the 49 cities? Clarify it in the manuscript

Answer: Good suggestion and thanks. Rainfall nitrogen concentration data is really from literatures. Due to the lack of data on nitrogen deposition in many cities, an average value was used for all cities. We have made a note of this in the revised manuscript (in lines of 166-170).

5、In Figure 8, why are 5 clusters generated to do this analysis rather than using all 49 cities? What do these 5 clusters stand for?

Answer: Good suggestions. The clustering algorithm is an unsupervised method, and a reasonable number of clusters needs to be set artificially. Of course, a reasonable number of clusters should be more effective and vivid in characterizing the study results. We found that the ANSA index for the study area has a clear hierarchical character (Figure 6), which prompted us to cluster 49 cities into five clusters. Each cluster, in effect, represents a cluster of cities with different gradients of agricultural indicators. This will be very effective to reveal the impact of agricultural restructuring on NANI.

6、Section 4.3. Please give some more specific suggestions on nitrogen management based on the findings of this study and delete those unrelated suggestions. For example, lines 391-393, domestic sewage and industrial wastewater are not analyzed in this manuscript, so that this suggestion is not supported by the results here.

Answer: Good suggestions and thanks. We have reorganized the section 4.3 to remove the reference to domestic sewage and industrial wastewater (in lines of 398-410).

Reviewer 2 Report

In this ms the authors aim to study the impact of agricultural structure adjustment on the NANI over the past 30 years (1990 - 2019) in Pearl River basin, a typical representative of the dramatic changes in China’s agricultural industrial structure with marked regional differences. They have used statistical data from the Agricultural Statistical Yearbook, and the National Economic and Social Development Statistical Bulletin. All the results are based on the applilcation of several equations for the quantification of the NANI and NANI(ASNA). The changes in agricultural industry  of China over the past 30 years, is discused in this ms, althought they concluded that the NANI(ASNA) instead of NANI indicator is more applicable, since it is difficult to directly quantify the impact of changes in agricultural industry structure on NANI. 

It is an interesting ms and well presented. I am wondering if these equations have been used before. If yes, I'd like the authors to add references. Moreover, references should be added in L. 46: "Mechanistic models are complex in structure and require many localized parameters" and L.: "....decreased by 78.5% and 76.9% in Fujian and Guangdong provinces, respectively". 

In my opinion, the ms is well structured and can be published with minor changes.

Author Response

Dear reviewers,

    Thank you for your letter and comments concerning our manuscript entitled “Effect of agricultural structure adjustment on spatial-temporal patterns of net anthropogenic nitrogen inputs in the Pearl River Basin from 1990 to 2019” (Manuscript ID: land-2095581). Your comments have proved very helpful with regard to our revisions and improvements of our paper, as well as providing insightful guidance for our research. We have gone through your comments carefully and have made necessary corrections. We have tried our best to improve the manuscript and we have made many changes to the text according to the comments of reviewers. All changes have been highlighted in red in the revised manuscript.

We sincerely appreciate the effort and consideration of the Editor/Reviewers and we hope that the corrections will meet with your approval. Once again, we thank you very much for your insightful comments and suggestions. Our point-to-point response to the comments is as follows:

 1、It is an interesting ms and well presented. I am wondering if these equations have been used before. If yes, I'd like the authors to add references. Moreover, references should be added in L. 46: "Mechanistic models are complex in structure and require many localized parameters" and L.: "....decreased by 78.5% and 76.9% in Fujian and Guangdong provinces, respectively".

Answer: Good suggestions and thanks. In the manuscript, equations (1-2) do appear in earlier literature and we have added references in the revised manuscript. We have also added the reference in the line of 46. The data on urbanisation ratios in Fujian and Guangdong provinces are derived from the official statistical yearbooks, not from the references. Therefore, we have not added references.

Reviewer 3 Report

The manuscript can be accepted as written.

Author Response

Dear reviewers,

Thank you for your letter and comments concerning our manuscript entitled “Effect of agricultural structure adjustment on spatial-temporal patterns of net anthropogenic nitrogen inputs in the Pearl River Basin from 1990 to 2019” (Manuscript ID: land-2095581). Your comments have proved very helpful with regard to our revisions and improvements of our paper, as well as providing insightful guidance for our research. We have gone through your comments carefully and have made necessary corrections. We have tried our best to improve the manuscript and we have made many changes to the text according to the comments of reviewers. All changes have been highlighted in red in the revised manuscript. 

1、The manuscript can be accepted as written.

Answer: Many thanks for the appreciation and praise for our works.
